# Optimal Location of Exit Doors for Efficient Evacuation of Crowds at Gathering Places

**Lino J. Alvarez-Vázquez** [1,*,†] , **Néstor García-Chan** [2,†] , **Aurea Martínez** [1,†] , **Carmen Rodríguez** [3,†] and **Miguel E. Vázquez-Méndez** [4,5,†]

1    Departamento Matemática Aplicada II, EI Telecomunicación, Universidade de Vigo, 36310 Vigo, Spain
2    Departamento Física, CUCEI, Universidad de Guadalajara, 44430 Guadalajara, Mexico
3    Departamento Matemática Aplicada, Facultad Matemáticas, Universidade de Santiago de Compostela, 15782 Santiago, Spain
4    CITMAGA, 15782 Santiago de Compostela, Spain
5    Departamento Matemática Aplicada, EPSE, Universidade de Santiago de Compostela, 27002 Lugo, Spain
*    Correspondence: lino@dma.uvigo.es
†    These authors contributed equally to this work.

**Abstract:** This work deals with the optimal design for the location of the exit doors at meeting places (such as sports centers, public squares, street markets, transport stations, etc.) to guarantee a safer emergency evacuation in events of a sporting, social, entertainment or religious type. This problem is stated as an optimal control problem of nonlinear partial differential equations, where the state system is a reformulation of the Hughes model (coupling the eikonal equation for a density-weighted walking velocity of pedestrians and the continuity equation for conservation of the pedestrian density), the control is the location of the exit doors at the domain boundary (subject to several geometric constraints), and the cost function is related to the evacuation rate. We provide a full numerical algorithm for solving the problem (a finite element technique for the discretization and a gradient-free procedure for the optimization), and show several numerical results for a realistic case.

**Keywords:** optimal location; exit doors; crowd evacuation; optimal control problem; mathematical modelling; simulation-based optimization

## 1. Introduction

Mass gatherings occur at a large number of venues for very different motivations, such as concerts, sports, religious events, political meetings, etc. Unfortunately, serious accidents continue to happen throughout the years at these places due to overcrowding [1].

Effective crowd evacuation in emergencies—for instance, fire, earthquake, or a terrorist attack—is a key public safety priority and, consequently, it is attracting considerate efforts to manage pedestrian evacuation in all types of emergencies. This is a very serious problem that can lead to injuries and even fatalities as a consequence of uncontrolled stampedes.

Modelling and analysis of crowd dynamics have been a very active study area in traffic engineering in recent decades. Despite the fact that crowd modeling started five decades ago with the pioneering works of Henderson [2] or Hirai and Tarui [3], most aspects of the recent studies are based on the systematic framework introduced in the seminal work of Hughes [4], a continuum-space model where the movement of a large crowd of pedestrians is modeled as a continuum medium, with a velocity computed by means of a constitutive law. A very interesting and extensive overview and classification of the different approaches employed in the mathematical literature—including microscopic, mesoscopic, macroscopic, and even multiscale viewpoints, which provide us with, respectively, agent-based, kinetic, fluid-dynamics-based and hybrid models—can be found, for instance, in the surveys of Bellomo

and Dogbé [5] or Martinez-Gil et al. [6], and the books of Kachroo et al. [7], Cristiani et al. [8] or Maury and Faure [9]. Although a complete existence theory for the Hughes model is still an open question, some contributions to this topic have been already given. We can highlight here, for instance, the works of Amadori et al. [10] and Di Francesco et al. [11], where the existence and uniqueness results for a one-dimensional regularized version of the Hughes model are addressed. However, the attention devoted to the numerical approximation of the model has been much broader, with a wide range of approaches. For the interested reader, among the large variety of articles dedicated to the study—mainly from a numerical viewpoint, but some of them also include theoretical aspects—of pedestrian flow modeling, we can stand out the recent works of Xia et al. [12], Huang et al. [13], Piccoli and Tosin [14], Hoogendoorn et al. [15], Carrillo et al. [16], Carlini et al. [17], Aghamohammadi and Laval [18] or Cristiani et al. [19].

Nevertheless, optimal control and optimization of these evacuation processes have been much more sparsely addressed within the mathematical literature. To evaluate the efficiency of the process during emergencies, evacuation time is the quantity commonly used. Also, in the context of an evacuation, a basic behavior of pedestrians is assumed [20], which means that pedestrians choose their walking direction without caring what the others do. There are several issues that can be controlled to provide minimum time during evacuation processes of crowds in gathering places. Among the smaller number of papers devoted to optimizing evacuation efficiency, we can mention those related to the control of the environment [20], the optimization of strategies design [21,22], emergency routes [23], obstacle placement [24,25], corridor widths [26], or sizes and locations of doors [27–30].

In this paper we introduce a novel framework, based on optimal control techniques of partial differential equations, to automate the optimization of locations for a given number of exit doors at gathering places, so that the evacuation of crowds takes place in a safer and faster way. Contrary to most of the papers that have previously dealt with the issue of exit doors optimization—which mainly uses statistical-type reasonings—our proposal is based on the use of optimal control techniques for a formulation of the problem within a framework of partial differential equations. In particular, we consider a reformulation of the classical Hughes system with a suitable set of initial/boundary conditions for modeling the flow of pedestrians (characterizing their density and walking velocity), which constitutes the state system of the optimal control problem. The objective function to be optimized in our problem corresponds to minimizing the number of pedestrians left inside the place at the end of the evacuation process. Moreover, we also need to include some constraints on the control (the location of the exit doors) since not all possible door positions are admissible for geometric, organizational, or security reasons. To solve the constrained optimal control problem numerically, we propose a full discretization of the problem, with a space semi-discretization via the finite element method over a family of triangular meshes of the domain under study, and a time semi-discretization via the Euler algorithm. Finally, for the resulting discretized minimization problem, we propose its optimization by means of any derivative-free algorithm, due to the hard numerical difficulties involved in the possible computation of the gradients of the cost functional. In our case, we have chosen the classical Nelder-Mead algorithm and a controlled random search procedure (both will be detailed in the below sections).

This paper is organized as follows. The mathematical model proposed to simulate the evolution of pedestrian flow is given in Section 2.1, and the full details of the formulation of our problem under the structure of an optimal control problem are detailed in Section 2.2. A complete numerical algorithm to solve this problem—including the numerical discretization and the optimization process—can be found in Section 2.3. Several numerical examples, corresponding to two different scenarios for a real-world study case, are presented and discussed in Section 3, to assess the effectiveness of our approach. Finally, some concluding remarks are summarized in the last Section 4.

## 2. Materials and Methods

### 2.1. The State System

In this subsection, we present the 2D mathematical model used in the numerical resolution of the control problem. First, we denote by $\Omega \subset \mathbb{R}^2$ and $[0, T] \subset \mathbb{R}$ the meeting place and the time interval under study, respectively. Then, we recall the original model introduced by Hughes for the flow of pedestrians, which couples the eikonal system with the continuity equation:

$$\|\nabla \phi\| = \frac{1}{f(\rho)} \quad \text{in } \Omega \times (0, T), \tag{1}$$

$$u(\rho) = -f(\rho) \frac{\nabla \phi}{\|\nabla \phi\|} \quad \text{in } \Omega \times (0, T), \tag{2}$$

$$\frac{\partial \rho}{\partial t} + \nabla \cdot (\rho u(\rho)) = 0 \quad \text{in } \Omega \times (0, T), \tag{3}$$

that must be complemented with a suitable set of boundary conditions on $\Gamma = \partial\Omega$ and initial conditions at $t = 0$. Equation (3) guarantees conservation of the mass of pedestrians, whose density is represented by $\rho(x, t)$. Equation (2) gives the walking velocity $u(x, t)$ for pedestrians, where the direction is given by the normalized gradient of $\phi(x, t)$ and the speed is obtained by a fundamental diagram $f$, establishing the relationship between velocity $u$ (or flow $\rho u$) and density $\rho$ of pedestrians—the particular expression for the fundamental diagram employed in our case can be found in Section 3. Finally, by considering this fundamental diagram, Equation (1) represents walking difficulties for high-density cases.

Taking into account that (1) can be rewritten as

$$\|\nabla \phi\|^2 = \frac{1}{f^2(\rho)}, \tag{4}$$

and adding a Laplacian term to (4) to increase the stability in its numerical resolution, we reformulate Equation (1) as the nonlinear second-order partial differential equation:

$$\|\nabla \phi\|^2 - \epsilon_1^2 \Delta \phi = \frac{1}{f^2(\rho)}, \tag{5}$$

where, for a sufficiently small parameter $\epsilon_1 > 0$, the viscosity solution of (5) is a regularization of the original solution of (1).

Then, arguing similarly to the case of Schrödinger equations [31], we introduce the direct transformation

$$\psi = e^{-\frac{\phi}{\epsilon_1}}, \tag{6}$$

(whose inverse transformation reads: $\phi = -\epsilon_1 \ln(\psi)$), and derive the following equation equivalent to (1) (see more details, for instance, in [32,33]):

$$\frac{1}{f^2(\rho)} \psi - \epsilon_1^2 \Delta \psi = 0. \tag{7}$$

With respect to the boundary conditions, we have to consider the boundary $\Gamma$ divided into three different parts: $\Gamma = \Gamma_w \cup \Gamma_{in} \cup \Gamma_{out}$, where $\Gamma_w$ corresponds to the lateral wall, $\Gamma_{in}$ represents the entry doors, and $\Gamma_{out}$ stands for the exit doors. Hughes model imposes that $\phi$ must vanish on the exit doors, and that $u(\rho) \cdot n$ must be null on the rest of the boundary (with $n$ the outward unit normal vector to $\Gamma$). So, due to the transformation (6), we have to impose on $\psi$ the boundary conditions $\psi = 1$ on $\Gamma_{out} \times (0, T)$, and $\nabla \psi \cdot n = 0$ on $(\Gamma_w \cup \Gamma_{in}) \times (0, T)$.

Finally, to avoid numerical instabilities in Equations (2) and (3), we replace these equations with the regularized ones:

$$u(\rho) = -f(\rho)\frac{\nabla\phi}{\sqrt{\|\nabla\phi\|^2 + \epsilon_2^2}},\tag{8}$$

$$\frac{\partial\rho}{\partial t} + \nabla\cdot(\rho u(\rho)) - \epsilon_3\Delta\rho = 0,\tag{9}$$

where $\epsilon_2, \epsilon_3 > 0$ are sufficiently small parameters.

Thus, our state system will be the following reformulation of the Hughes model:

$$\frac{1}{f^2(\rho)}\psi - \epsilon_1^2\Delta\psi = 0 \quad \text{in } \Omega\times(0,T),\tag{10}$$

$$\nabla\psi\cdot n = 0 \quad \text{on } (\Gamma_w\cup\Gamma_{in})\times(0,T),\tag{11}$$

$$\psi = 1 \quad \text{on } \Gamma_{out}\times(0,T),\tag{12}$$

$$\phi = -\epsilon_1\ln(\psi) \quad \text{in } \Omega\times(0,T),\tag{13}$$

$$u(\rho) = -f(\rho)\frac{\nabla\phi}{\sqrt{\|\nabla\phi\|^2 + \epsilon_2^2}} \quad \text{in } \Omega\times(0,T),\tag{14}$$

$$\frac{\partial\rho}{\partial t} + \nabla\cdot(\rho u(\rho)) - \epsilon_3\Delta\rho = 0 \quad \text{in } \Omega\times(0,T),\tag{15}$$

$$\nabla\rho\cdot n = 0 \quad \text{on } \Gamma_w\times(0,T),\tag{16}$$

$$\rho = \rho_{in} \quad \text{on } \Gamma_{in}\times(0,T),\tag{17}$$

$$\rho(0) = \rho^0 \quad \text{in } \Omega.\tag{18}$$

Now, multiplying Equations (10) and (15) by suitable test functions $\omega$, integrating by parts in $\Omega$, applying Green's formula and taking into account the corresponding Dirichlet and Neumann boundary conditions (11), (12), (16) and (17), we arrive to a standard variational formulation of the state system (further details on this formulation can be found, for instance, in [12] or [34]). In this variational formulation, and in order to enhance the evacuation of pedestrians avoiding jams in the exit doors, we will replace the boundary term $\int_{\Gamma_{out}}(\rho u(\rho)\cdot n)\,\omega\,d\Gamma$ by the reinforcement term $\int_{\Gamma_{out}}\gamma_{out}(\rho u(\rho)\cdot n)\,\omega\,d\Gamma$, with $\gamma_{out}\geq 1$ a strengthening parameter.

### 2.2. The Optimal Control Problem

We will formulate here the control problem consisting of the characterization of the optimal locations of the exit doors (which must be placed into an admissible part $\Gamma_{ad}$ of the boundary $\Gamma$ of $\Omega$) in such a way that the evacuation of the mass of pedestrians gathered together in $\Omega$ can be carried out as quickly as possible. For simplicity, we consider fixed the width of the doors, but it could be also considered variable, after incorporating the necessary (not straightforward) changes.

With this purpose in mind, we will choose as the cost function to be minimized, the sum of the number of pedestrians remaining inside $\Omega$ at all times during the evacuation period $[0, T]$, that is,

$$J = \int_0^T\int_\Omega\rho(x,t)\,dx\,dt,\tag{19}$$

where $\rho$ represents the pedestrians' density, solution of the above variational formulation of the state system (10)–(18).

**Remark 1.** *We must note here that, instead of the chosen objective function $J$, we could have chosen alternative options as, for instance, the number of pedestrians remaining at the interior of $\Omega$ at final time $T$ (i.e., $J = \int_\Omega\rho(x,T)\,dx$), or even the weighted sum of the number of pedestrians at several*

*intermediate times $\tau_1, \tau_2, \ldots, \tau_p \in [0, T]$ (i.e., $J = \sum_{i=1}^{p} \alpha_i \int_\Omega \rho(x, \tau_i) \, dx$, with $\alpha_i$, $i = 1, \ldots, p$, suitable weighting parameters).*

Thus, the optimal control problem to be solved consists of finding the optimal locations of the exit doors, such that these locations are in the admissible part of the boundary $\Gamma_{ad}$ and minimizing the cost function $J$.

### 2.3. Numerical Resolution

In this section, we will present a computational approach to the numerical solution of the above optimal control problem. So, we will discretize the state system (10)–(18) using a standard finite element method, and we will compute the numerical approximation of the nonlinear optimization problem (resulting from the full space-time discretization of the control problem) by means of a derivative-free algorithm.

#### 2.3.1. Space-Time Discretization

For the time semi-discretization, we consider a natural number $N \in \mathbb{N}$, and define the time step $\Delta t = \frac{T}{N}$. Then, we take the discretized times $t^n = n \Delta t$, for $n = 0, \ldots, N$. So, we will discretize the time derivative of $\rho$ in (15) by the Euler explicit method:

$$\frac{\partial \rho}{\partial t}(\cdot, t^n) \simeq \frac{\rho(\cdot, t^n) - \rho(\cdot, t^{n-1})}{\Delta t}, \quad n = 1, \ldots, N. \tag{20}$$

**Remark 2.** *Regarding the choice of the Euler method, it is worthwhile remarking here that we have also tried other alternative, more efficient methods for the time semi-discretization (for instance, second- or third-order Runge-Kutta algorithms, like in [33] or [12]) with slightly better numerical results, but with a much stronger computational effort. Since, in our simulation-based optimization approach, we will have to solve the full state system for each one of the many iterations in the optimization procedure, we need an algorithm with the lowest computational load possible, and a Euler algorithm—although is only a first-order method—seems accurate enough for our control purposes.*

For the space semi-discretization of the domain $\Omega$, we consider a family of triangular meshes $\tau_h$ for the polygonal approximation $\Omega_h$ of $\Omega$, with characteristic size $h$, and associated to it the Lagrange finite element space $P_1$ corresponding to globally continuous, piecewise linear polynomials on $\Omega_h$.

**Remark 3.** *Again, among the great variety of possible methods of semi-discretization in space, we have decided to choose the simplest one that, despite its low order of accuracy, is sufficient for our optimization problem, and has a very low computational cost. In fact, this method has been successfully employed by the authors in other related studies concerning a similar model for traffic flow [35,36]. Finally, we must mention that we have also tried more sophisticated space semi-discretization methods (for instance, a discontinuous Galerkin method, like in [12]), but the very slight improvements in the approximation were not compensated by the excessive increase in computation time, which is one of the key points for the effective resolution of the optimal control problem.*

Thus, considering this space-time discretization, the above variational formulation of the state system can be rewritten as a large system of equations, whose solution $\rho_h^n(\cdot) \simeq \rho(\cdot, t^n)$, $n = 0, \ldots, N$, will be used to compute the value of the discretized version of the cost function $J$ defined by (19):

$$J_h = \sum_{n=1}^{N} \Delta t \sum_{\tau \in \tau_h} \int_\tau \rho_h^n(x) \, dx, \tag{21}$$

where the integral on each element $\tau$ of the mesh $\tau_h$ can be approximated by any quadrature formula (in our case, the composite trapezoidal rule). We must note here that the location of the exit doors (the control in our problem) enters the value of $J_h$ via the definition of the exit boundary $\Gamma_{out}$ in boundary condition (12).

**Remark 4.** *It is also worthwhile remarking here that the approximation $\rho_h^0$ corresponds exactly to the initial condition $\rho^0$ in (18), since:*

$$\rho_h^0(\cdot) = \rho(\cdot, t^0) = \rho(\cdot, 0) = \rho^0(\cdot). \tag{22}$$

2.3.2. Numerical Optimization

Once we have determined how to compute the value of the discretized cost function $J_h$ for any arbitrary location of the exit doors, we can proceed to the minimization of this function $J_h$.

However, since we are dealing with a control-constrained optimal control problem, we need previously to convert our constrained optimization problem into an unconstrained one by adding a penalty term $P_h$ to the discretized cost function $J_h$, where the penalty term $P_h$ corresponds to the compliance with constraints $\Gamma_{out} \subset \Gamma_{ad}$.

Now, to minimize this new cost function $F_h = J_h + P_h$, we propose the use of a derivative-free algorithm, due to the highly nonlinear character of the problem. In particular, we will use two alternative methods: the Nelder-Mead simplex algorithm [37], and a controlled random search procedure for global optimization [38]. Numerical results will be presented and discussed in the next section.

We must recall here that the Nelder-Mead algorithm is a heuristic direct-search method, only based on function comparison (with no use of function gradients), widely used in the resolution of nonlinear optimization problems, which builds a sequence of simplices (by means of reflections, contractions, expansions, and shrinks) originally intended to converge to a minimum point. On the other hand, the controlled random search procedure is a conceptually simple and easily programmed method (also gradient-free), based on the use of different trial point generation schemes, that is effective in searching for global minima of multimodal functions, with or without constraints.

### 3. Results and Discussion

In this section, we will present the numerical results obtained in our computational simulations for a real-world case posed in the main square (Plaza Liberación) of the city of Guadalajara (Mexico), whose satellite photo (Google Earth, 2022) can be seen in Figure 1. A particular example of one of the many finite element meshes of the square employed in our optimization process can be found in Figure 2. All our numerical simulations have been carried out with the open-source scientific software FreeFem++ [39] interfaced with the optimization packages NelderMead (for the Nelder-Mead simplex algorithm) and CRS2 (for the controlled random search with local mutation method).

The square Plaza Liberación has the shape of a rectangle of approximate dimensions 180 by 92 m and includes numerous green areas with restricted access to pedestrians. Guadalajara is the second largest metropolis in Mexico, with a population of more than five million people in its metropolitan area, and the square Plaza Liberación—located in the center of the city and surrounded by the city hall building, the cathedral, a theater, and other administrative buildings—is the usual setting for mass gatherings of all kinds: cultural, political, social, etc. Therefore, the accumulation of crowds inside the square is very frequent, which sometimes leads to problems with their safe evacuation.

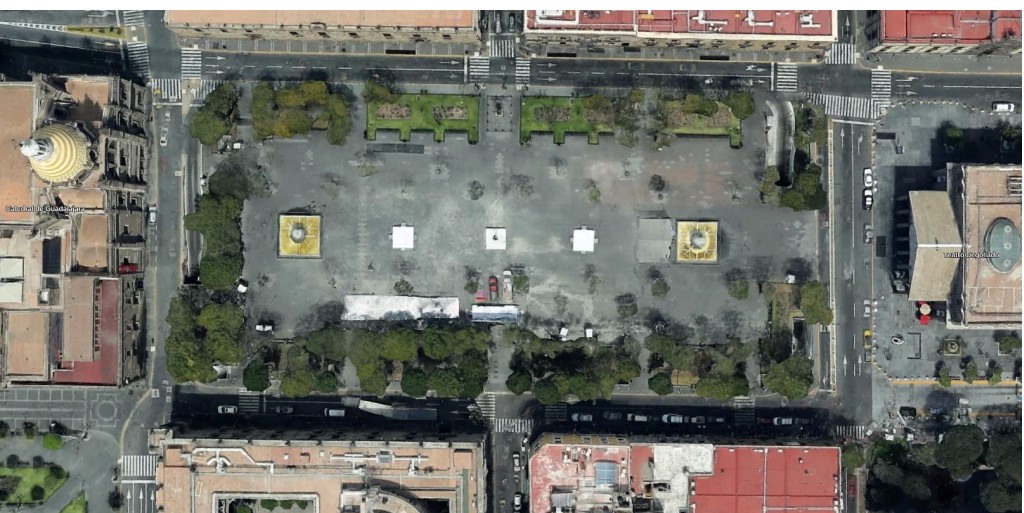

**Figure 1.** Satellite photo (Google Earth, 2022) of the domain $\Omega$ under study: Plaza Liberación in Guadalajara (Mexico), a rectangular square of approximately 180 by 92 m.

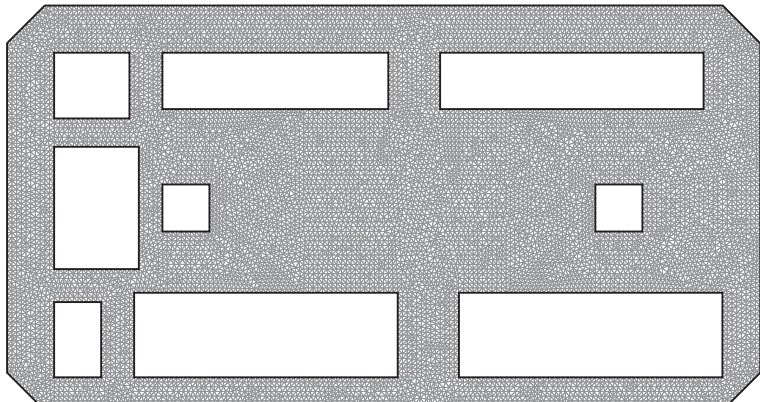

**Figure 2.** Example of a finite element mesh for the computational domain $\Omega_h$ used in the optimization process.

In our numerical experiences, for solving the state system coupling the eikonal equation with the continuity one we have chosen the following values for the parameters: $\epsilon_1 = 3.5$, $\epsilon_2 = 3.5$, and $\epsilon_3 = 0.001$, with a pedestrian walking speed $f(\rho)$ given by a fundamental diagram computed by the classical formula

$$f(\rho) = u_{max}\left(1 - \frac{\rho}{\rho_{max}}\right),\tag{23}$$

for a maximum velocity $u_{max} = 2$ m/s and a critical density $\rho_{max} = 10$ pedestrians/m². Furthermore, evacuation time $T$ is fixed to 350 s, with a time step of $\Delta t = 0.25$ (corresponding to a number of $N = 1400$ discretized times for the numerical resolution). For initial/boundary data we have considered an initial pedestrian density $\rho^0$ given, as can be seen in Figure 3, by a Gaussian distribution taking maximum value 5 in the central point of the square (where there is usually a pole waving the national flag), and a null boundary condition $\rho_{in} = 0$. Finally, to reinforce exit flux through doors, we have taken the strengthening parameter $\gamma_{out} = 1.2$.

Although we have performed a very large number of numerical experiences for very different scenarios, to show that our methodology is robust and efficient we will present only a few examples for the optimal location of two exit doors—with a given width of 4 m each one—for two different configurations of the admissible set $\Gamma_{ad}$. In the first case we want the exit doors to be located on the oblique sides at the bottom left and top left corners.

In the second configuration, the exit doors will be located on the longer left and right sides. (See Figure 4 for a clearer description of the admissible parts of the boundary in both cases).

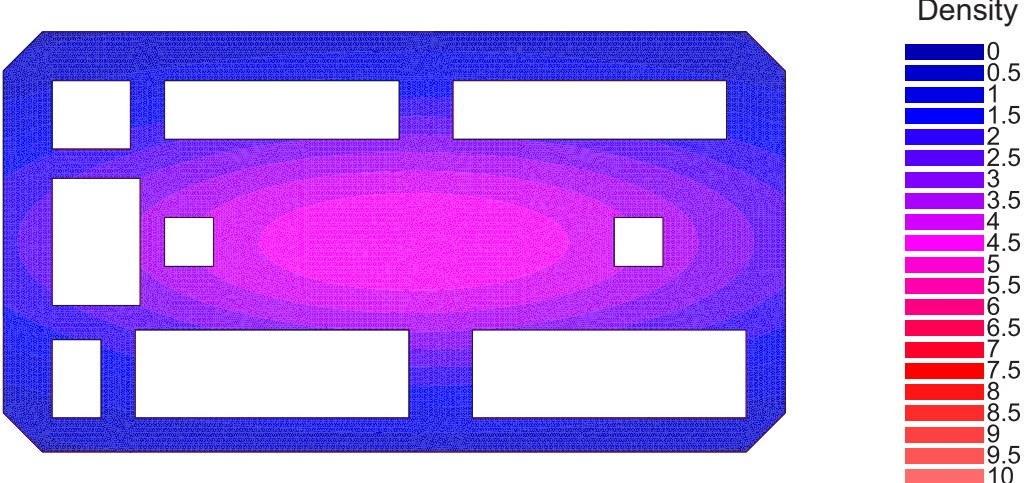

**Figure 3.** Initial density of pedestrians $\rho^0$ for the numerical tests: a Gaussian distribution taking maximum value 5 in the central point of the square.

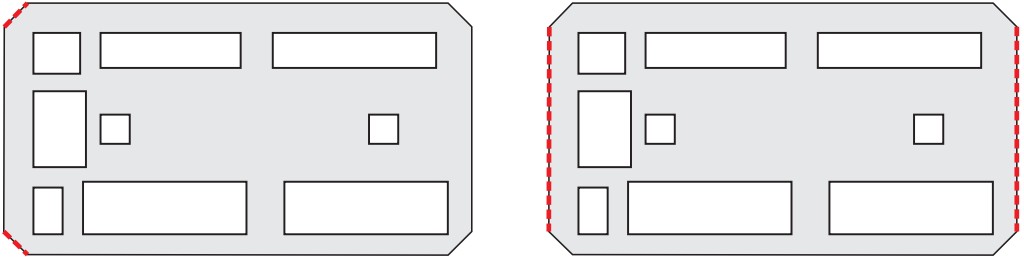

**Figure 4.** Admissible parts $\Gamma_{ad}$ (marked with dashed red lines) of the boundary $\Gamma$ where exit doors can be located for the first case (**left**) and for the second one (**right**).

*3.1. Case 1: Exit Doors in Left Corners*

For this first example, we have solved the optimal control problem by both optimization algorithms (the Nelder-Mead simplex algorithm and the controlled random search with the local mutation method). However, for the sake of brevity, we only present here the achieved solution for the former one, since this is the one that gives the best results in this case with a lower value of the cost function $F_h$.

For the numerical resolution of the state system, we have built triangular meshes $\tau_h$ of characteristic size $h = 1$. So, starting from a random location of the exit doors, corresponding to a cost function value of $F_h = 2.252 \times 10^7$, we arrive to the optimal location with a cost function value of $F_h = 1.804 \times 10^7$. In Figures 5 and 6 we can see, respectively, the normalized pedestrian velocities (for a normalized speed $\tilde{u} = \frac{f(\rho)}{u_{max}} = 1 - \frac{\rho}{\rho_{max}}$, with values ranging from 0 to 1) at final time $t = 350$ for the initial and the optimal locations (where the exit doors correspond to the outward arrows on the boundary), and the pedestrian densities $\rho$ at final time $t = 350$ for the initial and the optimal locations. It is worthwhile remarking here that with our achieved optimal configuration of the exit doors we get a complete evacuation of the square (depicted in Figure 6-bottom by a uniformly blue domain, corresponding to a constant null density of pedestrians $\rho = 0$), contrary to what happens with the initial random configuration (still presenting at the final time a high density of pedestrians in both left corners, as can be noticed in Figure 6-top).

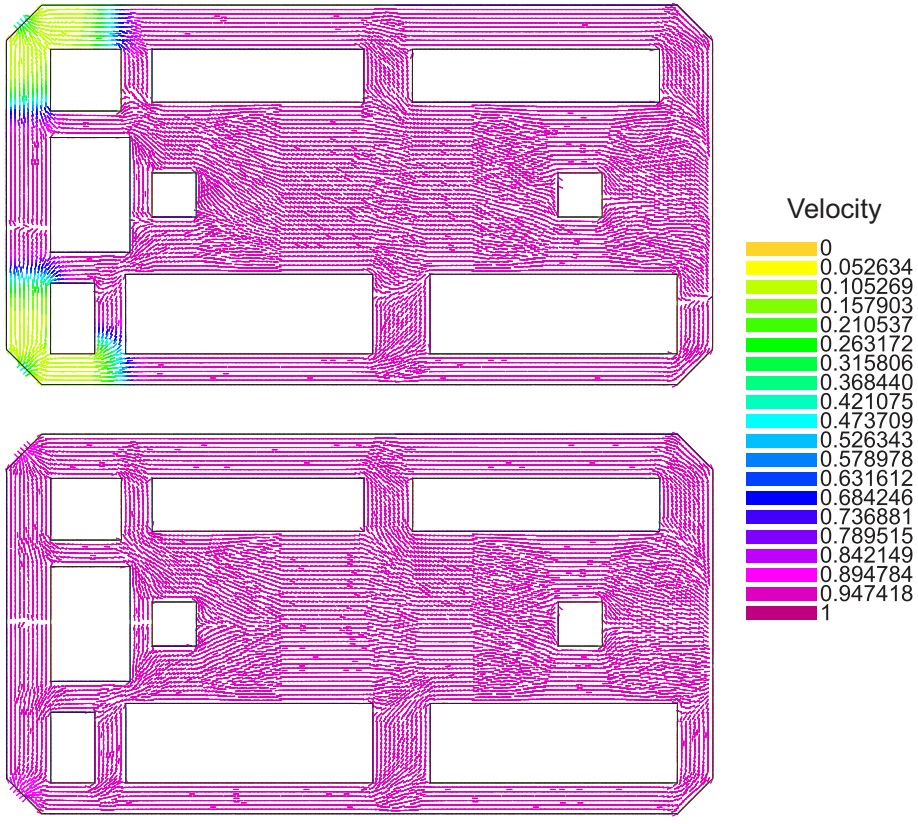

**Figure 5.** Case 1: Normalized pedestrian velocities $u/u_{max}$ at final time $t = 350$ for the initial (**top**) and the optimal locations (**bottom**) of exit doors (shown by outward arrows on the boundary).

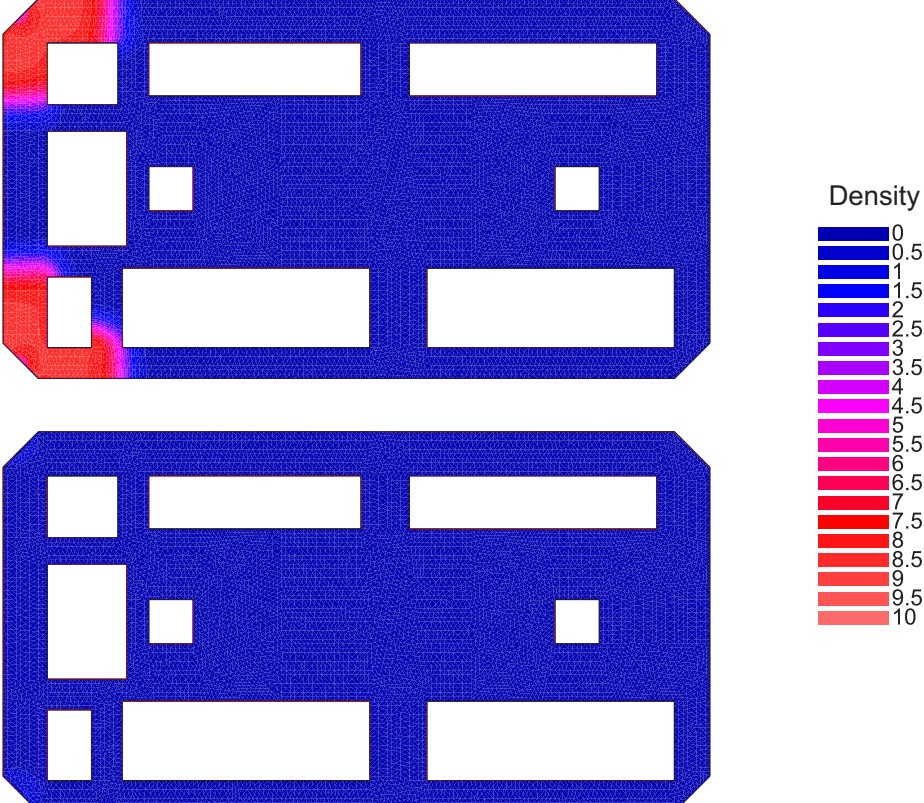

**Figure 6.** Case 1: Pedestrian densities $\rho$ at final time $t = 350$ for the initial (**top**) and the optimal locations (**bottom**).

We think that this example, where the admissible region $\Gamma_{ad}$ is small, is very relevant for our aims since it can be seen here how small changes in the locations of the doors can dramatically vary the results for the evacuation process, passing from a not fully evacuated place to a square completely empty (corresponding to the null density $\rho = 0$).

### 3.2. Case 2: Exit Doors in Left and Right Sides

In this second example, although we have also solved the optimal control problem by both optimization algorithms, we will present here only the optimal solution obtained by means of the controlled random search with the local mutation method, due to its better performance in this case—although withstanding a greater computational effort.

For the numerical resolution of the state system, we have worked now with finer triangular meshes $\tau_h$ of characteristic size $h = 0.5$. Thus, starting from a random location of the exit doors, corresponding to a cost function value of $F_h = 2.113 \times 10^7$, we arrive at the optimal location with a cost function value of $F_h = 1.891 \times 10^7$. Figures 7 and 8 represent, respectively, the normalized pedestrian velocities at final time $t = 350$ for the initial and the optimal locations (with exit doors shown again by outward arrows on the boundary), and the pedestrian densities at final time $t = 350$ for the initial and the optimal locations. We can see here that in this second case, as shown in Figure 8-bottom, the square cannot be completely evacuated within the set time, as a small number of pedestrians still need to exit the door on the right side. However, the pedestrian density here is appreciably lower than in the initial random case, as can be seen in Figure 8-top.

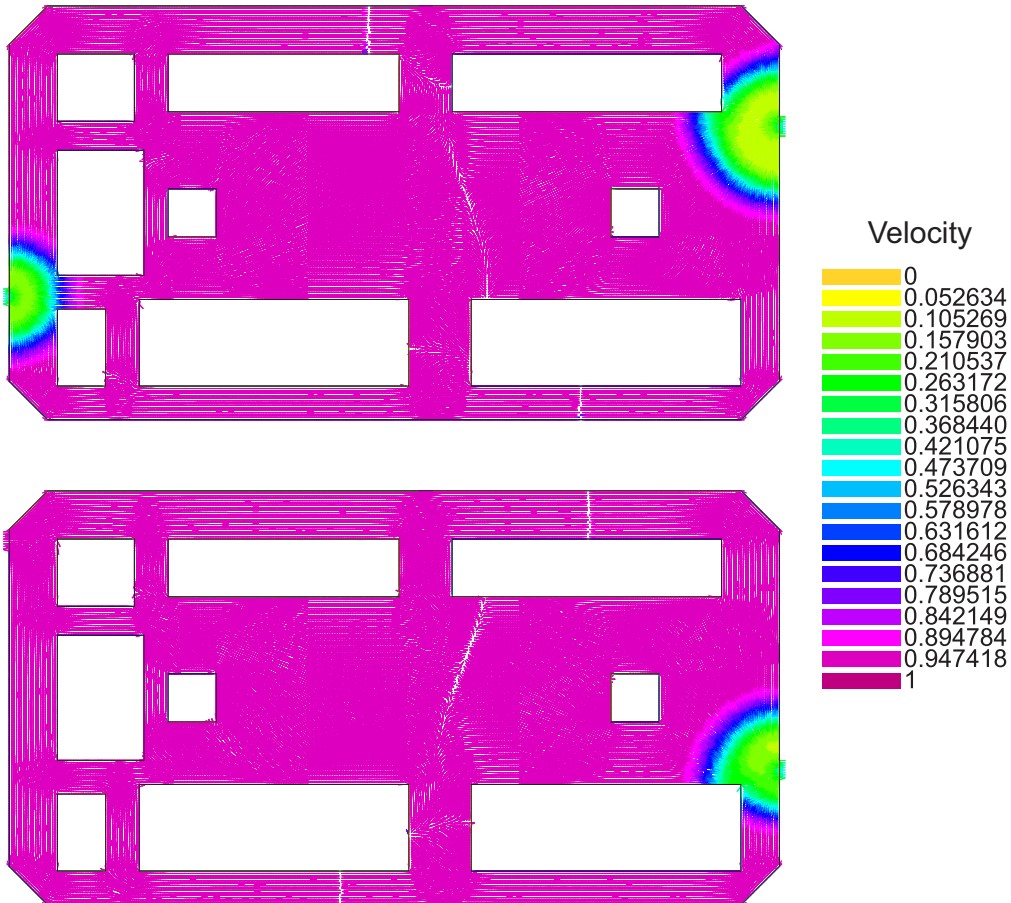

**Figure 7.** Case 2: Normalized pedestrian velocities $u/u_{max}$ at final time $t = 350$ for the initial (**top**) and the optimal locations (**bottom**) of exit doors (shown by outward arrows on the boundary).

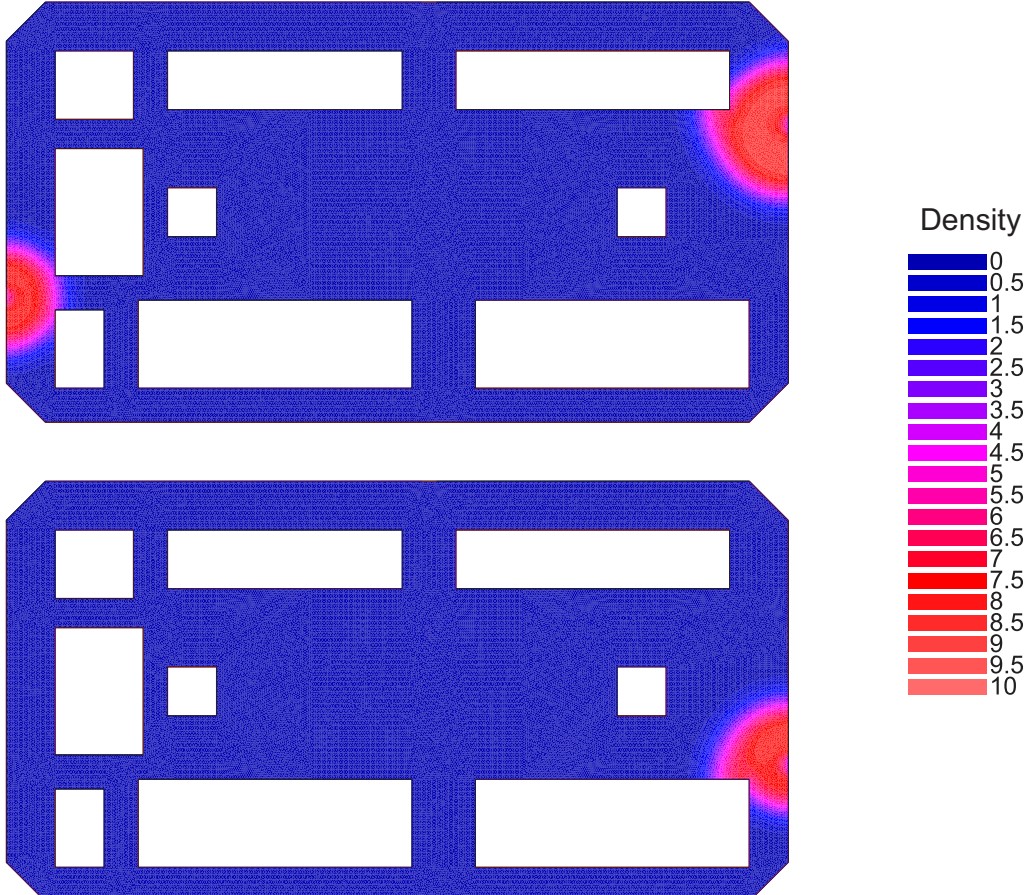

**Figure 8.** Case 2: Pedestrian densities $\rho$ at final time $t = 350$ for the initial (**top**) and the optimal locations (**bottom**).

Then, as a first consequence of the straightforward comparison of both above cases, we can conclude that the choice of exit doors in the left corners is a better option (in the sense of an easier evacuation) than doors located on the lateral sides.

## 4. Conclusions

In this paper, we propose a novel methodology for the optimal location of exit doors to assure an efficient evacuation of crowds at all types of gathering places. Our method is based on a regularized reformulation of the classical Hughes model for pedestrian flow (coupling the eikonal system for the walking velocity with the continuity equation for the density of pedestrians), where the associated optimal control problems are optimized by means of different derivative-free methods (in our particular cases, using the Nelder-Mead simplex algorithm and a controlled random search procedure for global optimization). Our original contribution to the topic consists of introducing an effective, systematic approach—based on the theory of optimal control of partial differential equations, within a simulation-based optimization framework—to the best choice of locations for exit doors in gathering places. Our methodology gives a scientifically based method to find the best options, complementing the traditionally employed use of the intuitive experience of stakeholders and decision-makers, and can be easily extended to other related design problems.

From the results obtained for the real-world study case shown in the above section, we can observe that our methodology shows good effectivity in identifying the optimal locations for the exit doors, allowing a safer and faster evacuation of pedestrians in highly crowded places. In addition, the robust structure of our method allows its use in a wide range of scenario variations: different numbers and configurations of exit doors, different

locations of admissible areas, the possibility of considering entry flows of pedestrians, the possible inclusion of temporary obstacles, etc.

**Author Contributions:** Writing, review and editing, L.J.A.-V., N.G.-C., A.M., C.R., and M.E.V.-M. All authors contributed equally. All authors have read and agreed to the published version of the manuscript.

**Funding:** This research was funded by Ministerio de Ciencia e Innovación (Spain) grant number TED2021-129324B-I00. N.G.-C. also thanks the support from Sistema Nacional de Investigadores (Mexico) under Grant SNI-52768, Programa para el Desarrollo Profesional Docente (Mexico) under Grant PRODEP/103.5/16/8066, and CONACyT by Ciencia de Frontera (Mexico) under Grant 217556.

**Data Availability Statement:** The authors confirm that the data supporting the findings of this study are available within the article.

**Conflicts of Interest:** The authors declare no conflict of interest. The funders had no role in the design of the study; in the collection, analyses, or interpretation of data; in the writing of the manuscript, or in the decision to publish the results.

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
