# Peer review of "Optimal Location of Exit Doors for Efficient Evacuation of Crowds at Gathering Places"

_axioms, doi:10.3390/axioms11110592_

Round 1

Reviewer 1 Report

I read this paper in detail. This paper should be revised via follows:

*"....Keywords: optimal location;..." should be typed as "..Keywords: Optimal location;..".

* [7–13] these papers should be written in detail.

*Eq4 should be ended with comma.

*Eq5 should be ended with comma.

*Eq6 should be ended with comma.

*Authors should pay attention to punctuation marks at the end of equations.

*"Figure 1" should be official.

*Eq23 should be ended with comma.

*Author should review the whole manuscript and correct all the grammatical and typo errors.

*Figures should be explained in detail.

*The authors are requested to add more details regarding their original contributions in this manuscript.

Reviewer 2 Report

I think that the paper is well written and go straight to the point. Also the topic is interesting and the real application is valuable. However, there are some important points which must imperatively revised.

Line 24: The state of the art must be rewritten completely. How can authors state that "all the studies on crowd dynamics take as a starting point the pioneering work of Hughes"? Studies on this topic began 30 years before!

I suggest having a look at the paper https://arxiv.org/abs/2207.08696 for a complete list of seminal papers as well as recent review papers. Moreover, in the paper https://arxiv.org/abs/2108.00086 authors can find some other references about the Hughes model.

Finally, if the authors find it useful, they can cite the paper https://epubs.siam.org/doi/10.1137/140962413 which uses similar techniques for optimizing pedestrian outflow.

Line 85: Give the definition of the fundamental diagram.

Line 104: please add a reference for this "standard variational formulation".

Line 109: It is better if you mention the fact that in your study the width of the door is fixed. Otherwise things get more complicated, especially beacuse of the fact that in macroscopic models granularity effects (1-to-1 interactions) are neglected.

Line 130. The numerical discretization of the equations is totally missing. Authors should add some materials and add references to prior work.

Line 152: please explain in a few words the main ideas behind the two optimization methods.

NUMERICAL TESTS: I like Test 2, but I do not like at all Test 1, for several reasons: Gamma_ad is too small, initial exits' locations are too close to the opimal ones, Fig.5-bottom is just the (-gradient of the) solution of the eikonal equation (since rho=0), Fig.6-bottom is just equal to Fig.2 coloured in blue. I think that authors should make an effort to find another relevant test.

MINORS:

1. eykonal -->eikonal (2 times)

2. for equation (23) I would say "classical formula" instead of "classical scheme".

3. Ref. [21], "I nfluence"

4. Ref. [22], missing first name of Shu

Reviewer 3 Report

The authors claim that they propose a novel methodology, however the emphasis of the manuscript is on the application of known algorithms on the space and time semidiscretization, and the subsequent optimization. No comparison to other methodologies in terms of efficiency, stability etc is presented. The paper contains 8 figures on the the application of the methodology to a specific problem, but lacks the the method development, analysis and comparison.

Please see below for other remarks:

The error analysis of the used algorithms should be present. Critical properties of all methods used must be presented, such as the order of accuracy, stability characteristics etc.

In lines 102-107, the derivation of the standard variational formulation is mentioned but not described in detail. The computations of the derivation are essential.

Why is the Euler used for the time semidiscretization? There are much more efficient methods.

In two instances "eykonal" has been used as opposed to "eikonal"

Round 2

Reviewer 2 Report

Revised version is fine to me

Reviewer 3 Report

Some minor corrections have been made. However, the emphasis of the manuscript is still on the application of the algorithm, and fails to show the development and analysis of the numerical method.